# Adversarially Robust Streaming Algorithms via Differential Privacy

**Avinatan Hassidim**[*]    **Haim Kaplan**[†]    **Yishay Mansour**[†]    **Yossi Matias**[‡]    **Uri Stemmer**[§]

## Abstract

A streaming algorithm is said to be *adversarially robust* if its accuracy guarantees are maintained even when the data stream is chosen maliciously, by an *adaptive adversary*. We establish a connection between adversarial robustness of streaming algorithms and the notion of *differential privacy*. This connection allows us to design new adversarially robust streaming algorithms that outperform the current state-of-the-art constructions for many interesting regimes of parameters.

## 1 Introduction

The field of *streaming algorithms* was formalized by Alon, Matias, and Szegedy [3], and has generated a large body of work that intersects many other fields in computer science such as theory, databases, networking, and natural language processing. Consider a scenario in which data items are being generated one by one, e.g., IP traffic monitoring or web searches. Generally speaking, streaming algorithms aim to process such data streams while using only a limited amount of memory, significantly smaller than what is needed to store the entire data stream.[5] Typical streaming problems include estimating frequency moments, counting the number of distinct elements in the stream, identifying heavy-hitters in the stream, estimating the median of the stream, and much more [22, 13, 4, 34, 16, 14, 15, 28, 37, 21, 31, 38].

Usually, streaming algorithms can be queried a lot of times throughout the execution. The reason is that (usually) the space requirement of streaming algorithms scales as $\log(1/\delta)$, where $\delta$ is the failure probability of the algorithm. By a union bound, this means that in order to guarantee accuracy for $m$ queries (with probability $1 - \delta$) the space only scales proportionally to $\log(m/\delta)$, so we can tolerate quite a few queries without blowing up space. However, for this argument to go through, we need to assume that the entire stream is *fixed* in advanced (and is just given to us one item at a time), or at least that the choice of the items in the stream is *independent* of the internal state (and coin tosses) of our algorithm. This setting is sometimes referred to as the *oblivious* setting. The vast majority of the work on streaming algorithms is focused on the oblivious setting.

Now suppose that the items in the stream, as well as the queries issued to the algorithm, are chosen by an *adaptive (stateful) adversary*. Specifically, every item in the stream (and each of the queries) is chosen by the adversary as a function of the previous items in the stream, the previous queries, and the previous answers given by our streaming algorithm. As a result, the items in the stream are *no longer independent* of the internal state of our algorithm. Oblivious streaming algorithms fail to provide meaningful utility guarantees in such a situation. In this work we aim to design *adversarially robust streaming algorithms* that maintain (provable) accuracy against such adaptive adversaries,

---

[*]Bar-Ilan University and Google.

[†]Tel Aviv University and Google.

[‡]Google.

[§]Ben-Gurion University and Google.

[5]We remark, however, that streaming algorithms are also useful in the offline world, for example in order to process a large unstructured database that is located on an external storage.

while of course keeping the memory and runtime requirements to a minimum. We stress that such dependencies between the items in the stream and the internal state of the algorithm may occur unintentionally (even when there is no "adversary"). For example, consider a large system in which a streaming algorithm is used to analyze data coming from one part of the system while answering queries generated by another part of the system, but these (supposedly) different parts of the system are connected via a feedback loop. In such a case, it is no longer true that the items in the stream are generated independently of the previous answers, and the vast majority of the existing streaming algorithms would fail to provide meaningful utility guarantees.

Recall that (typically) in the *oblivious setting* the memory requirement only grows logarithmically with the number $m$ of queries that we want to support. For the *adaptive setting*, one can easily show that a memory blowup of $\tilde{O}(m)$ suffices. This can be achieved, e.g., by running $m$ independent copies of the algorithm (where we feed the input stream to each of the copies) and using each copy in order to answer at most one query. Can we do better?

This question has motivated a recent line of work that is focused on constructing *adversarially robust streaming algorithms* [36, 23, 24, 1, 2, 27, 9, 8]. The formal model we consider was recently put forward by Ben-Eliezer et al. [8], who presented adversarially robust streaming algorithms for many problems in the *insertion-only model* (i.e., when the stream contains only *positive* updates). Moreover, their results extend to *turnstile streams* (where both *positive* and *negative* updates are allowed), provided that the number of negative updates is small. The question remained largely open for the general turnstile model where there might be a large number of negative updates.

## 1.1 Existing Results

We now give an informal overview of the techniques of [8]. This intuitive overview is generally oversimplified, and hides many of the difficulties that arise in the actual analysis. See [8] for the formal details and for additional results.

Consider a stream of updates $(a_1, \Delta_1), \ldots, (a_m, \Delta_m)$, where $a_i \in [n]$ is the $i$th element and $\Delta_i \in \mathbb{Z}$ is its weight. For $i \in [m]$ we write $\vec{a}_i = ((a_1, \Delta_1), \ldots, (a_i, \Delta_i))$ to denote the first $i$ updates in the stream. Let $g : ([n] \times \mathbb{Z})^* \to \mathbb{R}$ be a function (for example, $g$ might count the number of distinct elements in the stream). At every time step $i$, after obtaining the next update in the stream $(a_i, \Delta_i)$, our goal is to output an approximation for $g(\vec{a}_i)$.

Ben-Eliezer et al. [8] focused on the case where all of the weights $\Delta_i$ are *positive* (this assumption is known as the *insertion-only model*). To illustrate the results of [8], let us consider the *distinct elements* problem, in which the function $g$ counts the number of distinct elements in the stream. Specifically, after every update $(a_i, \Delta_i)$ we need to output an estimation of $g(\vec{a}_i) = |\{a_j : j \in [i]\}|$. Observe that, in the insertion-only model, this quantity is monotonically increasing. Furthermore, since we are aiming for a multiplicative $(1 \pm \alpha)$ error, even though the stream is large (of length $m$), the number of times we actually need to modify the estimates we release is quite small (roughly $\frac{1}{\alpha} \log m$ times). Informally, the idea of [8] is to run several independent sketches in parallel, and to use each sketch to release answers over a part of the stream during which the estimate remains constant. In more detail, the generic transformation of [8] (applicable not only to the distinct elements problem) is based on the following definition.

**Definition 1.1** (Flip number [8]). *Given a function $g$, the $(\alpha, m)$-flip number of $g$, denoted as $\lambda_{\alpha,m}(g)$, is the maximal number of times that the value of $g$ can change (increase or decrease) by a factor of $(1 + \alpha)$ during a stream of length $m$.*

The generic construction of [8] for a function $g$ is as follows.

1. Instantiate $\lambda \geq \lambda_{\alpha,m}(g)$ independent copies of an oblivious streaming algorithm for the function $g$, and set $j = 1$.

2. When the next update $(a_i, \Delta_i)$ arrives:
   (a) Feed $(a_i, \Delta_i)$ to *all* of the $\lambda$ copies.
   (b) Release an estimate using the $j$th copy (rounded to the nearest power of $(1 + \alpha)$). If this estimate is different than the previous estimate, then set $j \leftarrow j + 1$.

Ben-Eliezer et al. [8] showed that this can be used to transform an oblivious streaming algorithm for $g$ into an adversarially robust streaming algorithm for $g$. In addition, the overhead in terms of memory

is only $\lambda_{\alpha,m}(g)$, which is typically small in the insertion-only model (typically $\lambda_{\alpha,m}(g) \lesssim \frac{1}{\alpha} \log m$). Moreover, [8] showed that their techniques extend to the *turnstile model* (when the stream might contain updates with negative weights), provided that the number of negative updates is small (and so $\lambda_{\alpha,m}(g)$ remains small).

**Theorem 1.2** ([8], informal). *Fix any function $g$ and let $\mathcal{A}$ be an oblivious streaming algorithm for $g$ that for any $\alpha, \delta > 0$ uses space $L(\alpha, \delta)$ and guarantees accuracy $\alpha$ with success probability $1 - \delta$ for streams of length $m$. Then there exists an adversarially robust streaming algorithm for $g$ that guarantees accuracy $\alpha$ with success probability $1 - \delta$ for streams of length $m$ using space*

$$O\left(L\left(\frac{\alpha}{10}, \delta\right) \cdot \lambda_{\frac{\alpha}{10}, m}(g)\right).$$

## 1.2 Our Results

We establish a connection between adversarial robustness of streaming algorithms and *differential privacy*, a model to provably guarantee privacy protection when analyzing data. Consider a database containing (sensitive) information pertaining to individuals. An algorithm operating on such a database is said to be *differentially private* if its outcome does not reveal information that is specific to any individual in the database. More formally, differential privacy requires that no individual's data has a significant effect on the distribution of the output. Intuitively, this guarantees that whatever is learned about an individual could also be learned with her data arbitrarily modified (or without her data). Formally,

**Definition 1.3** ([18]). *Let $\mathcal{A}$ be a randomized algorithm that operates on databases. Algorithm $\mathcal{A}$ is $(\varepsilon, \delta)$-differentially private if for any two databases $S, S'$ that differ on one row, and any event $T$, we have*

$$\Pr[\mathcal{A}(S) \in T] \leq e^{\varepsilon} \cdot \Pr[\mathcal{A}(S') \in T] + \delta.$$

Our main conceptual contribution is to show that the notion of differential privacy can be used *as a tool* in order to construct new adversarially robust streaming algorithms. In a nutshell, the idea is to protect the *internal state* of the algorithm using *differential privacy*. Loosely speaking, this limits (in a precise way) the dependency between the internal state of the algorithm and the choice for the items in the stream, and allows us to analyze the utility guarantees of the algorithm even in the adaptive setting. Notice that differential privacy is *not* used here in order to protect the privacy of the data items in the stream. Rather, differential privacy is used here to protect the internal randomness of the algorithm.

For many problems of interest, even in the general turnstile model (with deletions), this technique allows us to obtain adversarially robust streaming algorithms with sublinear space. To the best of our knowledge, our technique is the first to provide meaningful results for the general turnstile model. In addition, for interesting regimes of parameters, our algorithm outperforms the current state-of-the-art constructions also for the insertion-only model (strictly speaking, our results for the insertion-only model are incomparable with [8]).

We obtain the following theorem.

**Theorem 1.4.** *Fix any function $g$ and fix $\alpha, \delta > 0$. Let $\mathcal{A}$ be an oblivious streaming algorithm for $g$ that uses space $L\left(\frac{\alpha}{10}, \frac{1}{10}\right)$ and guarantees accuracy $\frac{\alpha}{10}$ with success probability $\frac{9}{10}$ for streams of length $m$. Then there exists an adversarially robust streaming algorithm for $g$ that guarantees accuracy $\alpha$ with success probability $1 - \delta$ for streams of length $m$ using space*

$$O\left(L\left(\frac{\alpha}{10}, \frac{1}{10}\right) \cdot \sqrt{\lambda_{\frac{\alpha}{10}, m}(g) \cdot \log\left(\frac{1}{\delta}\right)} \cdot \log\left(\frac{m}{\alpha\delta}\right)\right).$$

Compared to [8], our space bound grows only as $\sqrt{\lambda}$ instead of linearly in $\lambda$. This means that in the general turnstile model, when $\lambda$ can be large, we obtain a significant improvement at the cost of additional logarithmic factors. In addition, as $\lambda$ typically scales at least linearly with $1/\alpha$, we obtain improved bounds even for the insertion-only model in terms of the dependency of the memory in $1/\alpha$ (again, at the expense of additional logarithmic factors).

## 1.3 Other Related Results

Over the last few years, differential privacy has proven itself to be an important *algorithmic* notion (even when data privacy is not of concern), and has found itself useful in many other fields, such as machine learning, mechanism design, secure computation, probability theory, secure storage, and more. [35, 17, 26, 41, 5, 39, 40, 33, 6] In particular, our results utilize a connection between *differential privacy* and *generalization*, which was first discovered by Dwork et al. [17] in the context of *adaptive data analysis*.

# 2 Preliminaries

A stream of length $m$ over a domain $[n]$ consists of a sequence of updates $(a_1, \Delta_1), \dots, (a_m, \Delta_m)$ where $a_i \in [n]$ and $\Delta_i \in \mathbb{Z}$. For $i \in [m]$ we write $\vec{a}_i = ((a_1, \Delta_1), \dots, (a_i, \Delta_i))$ to denote the first $i$ updates in the stream. Let $g : ([n] \times \mathbb{Z})^* \to \mathbb{R}$ be a function (for example, $g$ might count the number of distinct elements in the stream). At every time step $i$, after obtaining the next update in the stream $(a_i, \Delta_i)$, our goal is to output an approximation for $g(\vec{a}_i)$. We assume throughout the paper that $\log(m) = \Theta(\log n)$ and that $g$ is bounded polynomially in $n$.

## 2.1 Streaming against adaptive adversary

The adversarial streaming model, in various forms, was considered by [36, 23, 24, 1, 2, 27, 9, 8]. We give here the formulation presented by Ben-Eliezer et al. [8]. The adversarial setting is modeled by a two-player game between a (randomized) `StreamingAlgorithm` and an `Adversary`. At the beginning, we fix a function $g$. Then the game proceeds in rounds, where in the $i$th round:

1. The `Adversary` chooses an update $u_i = (a_i, \Delta_i)$ for the stream, which can depend, in particular, on all previous stream updates and outputs of `StreamingAlgorithm`.

2. The `StreamingAlgorithm` processes the new update $u_i$ and outputs its current response $z_i$.

The goal of the `Adversary` is to make the `StreamingAlgorithm` output an incorrect response $z_i$ at some point $i$ in the stream. For example, in the distinct elements problem, the adversary's goal is that at some step $i$, the estimate $z_i$ will fail to be a $(1 + \alpha)$-approximation of the true current number of distinct elements.

We remark that our techniques extend to a model in which the `StreamingAlgorithm` only needs to release an approximation for $g(\vec{a}_i)$ in at most $w \leq m$ time steps (which are chosen adaptively by the adversary), in exchange for lower space requirements. For simplicity, we will focus on the case where the `StreamingAlgorithm` needs to release an approximate answer in every time step.

## 2.2 Preliminaries from differential privacy

**The Laplace Mechanism.** The most basic constructions of differentially private algorithms are via the Laplace mechanism as follows.

**Definition 2.1** (The Laplace distribution). *A random variable has probability distribution* $\mathrm{Lap}(b)$ *if its probability density function is* $f(x) = \frac{1}{2b} \exp\left(-\frac{|x|}{b}\right)$, *where* $x \in \mathbb{R}$.

**Definition 2.2** (Sensitivity). *A function* $f : X^* \to \mathbb{R}$ *has* sensitivity $\ell$ *if for every two databases* $S, S' \in X^*$ *that differ in one row it holds that* $|f(S) - f(S')| \leq \ell$.

**Theorem 2.3** (The Laplace mechanism [18]). *Let* $f : X^* \to \mathbb{R}$ *be a sensitivity* $\ell$ *function. The mechanism that on input* $S \in X^*$ *returns* $f(S) + \mathrm{Lap}(\frac{\ell}{\varepsilon})$ *preserves* $(\varepsilon, 0)$-*differential privacy.*

**Example 2.4.** *Consider a database* $S$ *containing the medical records of* $n$ *individuals. Suppose that we are interested in privately estimating the number of individuals with diabetes, and let* $f(S)$ *denote this number. Observe that the sensitivity of* $f$ *is 1, since modifying one record in the data can change the number of individuals with diabetes by at most 1. Therefore, Theorem 2.3 states that we can privately estimate* $f(S)$ *by adding noise sampled from* $\mathrm{Lap}(\frac{1}{\varepsilon})$. *Observe that the noise magnitude is independent of the database size. Hence, when the database* $S$ *is large, the noise we add for privacy has only a very small (relative) effect on the result.*

**The sparse vector technique.**  Consider a large number of low-sensitivity functions $f_1, f_2, \ldots$ which are given (one by one) to a data curator (holding a database $S$). Dwork, Naor, Reingold, Rothblum, and Vadhan [19] presented a simple (and elegant) tool that can *privately* identify the first index $i$ such that the value of $f_i(S)$ is "large".

---

**Algorithm AboveThreshold**
**Input:** Database $S \in X^*$, privacy parameter $\varepsilon$, threshold $t$, and a stream of sensitivity-1 queries $f_i : X^* \to \mathbb{R}$.

  1. Let $\hat{t} \leftarrow t + \mathrm{Lap}(\frac{2}{\varepsilon})$.

  2. In each round $i$, when receiving a query $f_i$, do the following:

     (a) Let $\hat{f}_i \leftarrow f_i(S) + \mathrm{Lap}(\frac{4}{\varepsilon})$.

     (b) If $\hat{f}_i \geq \hat{t}$, then output $\top$ and halt.

     (c) Otherwise, output $\bot$ and proceed to the next iteration.

---

Notice that the number of possible rounds unbounded. Nevertheless, this process preserves differential privacy:

**Theorem 2.5** ([19, 25]). *Algorithm* AboveThreshold *is $(\varepsilon, 0)$-differentially private.*

**Privately approximating the median of the data.**  Given a database $S \in X^*$, consider the task of *privately* identifying an *approximate median* of $S$. Specifically, for an error parameter $\Gamma$, we want to identify an element $x \in X$ such that there are at least $|S|/2 - \Gamma$ elements in $S$ that are bigger or equal to $x$, and there are at least $|S|/2 - \Gamma$ elements in $S$ that are smaller or equal to $x$. The goal is to keep $\Gamma$ as small as possible, as a function of the privacy parameters $\varepsilon, \delta$, the database size $|S|$, and the domain size $|X|$.

There are several advanced constructions in the literature with error that grows very slowly as a function of the domain size (only polynomially with $\log^* |X|$). [7, 12, 11, 32] In our application, however, the domain size is already small, and hence, we can use simpler constructions (where the error grows logarithmically with the domain size).

**Theorem 2.6.** *There exists an $(\varepsilon, 0)$-differentially private algorithm that given a database $S \in X^*$ outputs an element $x \in X$ such that with probability at least $1 - \delta$ there are at least $|S|/2 - \Gamma$ elements in $S$ that are bigger or equal to $x$, and there are at least $|S|/2 - \Gamma$ elements in $S$ that are smaller or equal to $x$, where $\Gamma = O\left(\frac{1}{\varepsilon} \log\left(\frac{|X|}{\delta}\right)\right)$.*

**Composition of differential privacy.**  The following theorem allows to argue about the privacy guarantees of an algorithm that accesses its input database using several differentially private mechanisms.

**Theorem 2.7** ([20]). *Let $0 < \varepsilon, \delta' \leq 1$, and let $\delta \in [0, 1]$. A mechanism that permits $k$ adaptive interactions with mechanisms that preserves $(\varepsilon, \delta)$-differential privacy (and does not access the database otherwise) ensures $(\varepsilon', k\delta + \delta')$-differential privacy, for $\varepsilon' = \sqrt{2k \ln(1/\delta')} \cdot \varepsilon + 2k\varepsilon^2$.*

**Generalization properties of differential privacy.**  Dwork et al. [17] and Bassily et al. [5] showed that if a predicate $h$ is the result of a differentially private computation on a random sample, then the empirical average of $h$ and its expectation over the underlying distribution are guaranteed to be close.

**Theorem 2.8** ([17, 5]). *Let $\varepsilon \in (0, 1/3)$, $\delta \in (0, \varepsilon/4)$, and $n \geq \frac{1}{\varepsilon^2} \log(\frac{2\varepsilon}{\delta})$. Let $\mathcal{A} : X^n \to 2^X$ be an $(\varepsilon, \delta)$-differentially private algorithm that operates on a database of size $n$ and outputs a predicate $h : X \to \{0, 1\}$. Let $\mathcal{D}$ be a distribution over $X$, let $S$ be a database containing $n$ i.i.d. elements from $\mathcal{D}$, and let $h \leftarrow \mathcal{A}(S)$. Then*

$$\Pr_{\substack{S \sim \mathcal{D}^n \\ h \leftarrow \mathcal{A}(S)}} \left[ \left| \frac{1}{|S|} \sum_{x \in S} h(x) - \mathbb{E}_{x \sim \mathcal{D}}[h(x)] \right| \geq 10\varepsilon \right] < \frac{\delta}{\varepsilon}.$$

---

**Algorithm 1** `RobustSketch`

---

**Input:** Parameters $\alpha, \lambda, \delta, k$, and a collection of $k$ random strings $R = (r_1, \ldots, r_k) \in (\{0,1\}^*)^k$.
**Algorithm used:** An oblivious streaming algorithm $\mathcal{A}$ for a functionality $g$ that guarantees that with probability at least $9/10$, all its estimates are accurate to within multiplicative error of $(1 \pm \frac{\alpha}{10})$.

1. Initialize $k$ independent instances $\mathcal{A}_1, \ldots, \mathcal{A}_k$ of algorithm $\mathcal{A}$ with the random strings $r_1, \ldots, r_k$, respectively.

2. Let $\tilde{g} \leftarrow g(\perp)$ and denote $\varepsilon = \frac{1}{100}$ and $\varepsilon_0 = \frac{\varepsilon}{16\sqrt{\lambda \ln(1/\delta)}}$

3. REPEAT at most $\lambda$ times (outer loop)

    (a) Let $\hat{t} \leftarrow \frac{k}{2} + \text{Lap}(\frac{1}{\varepsilon_0})$

    (b) REPEAT (inner loop)
        i. Receive next update $(a_i, \Delta_i)$
        ii. Insert update $(a_i, \Delta_i)$ into each algorithm $\mathcal{A}_1, \ldots, \mathcal{A}_k$ and obtain answers $y_{i,1}, \ldots, y_{i,k}$
        iii. If $\left| \left\{ j : \tilde{g} \notin (1 \pm \frac{\alpha}{2}) \cdot y_{i,j} \right\} \right| + \text{Lap}(\frac{1}{\varepsilon_0}) < \hat{t}$, then output estimate $\tilde{g}$ and CONTINUE inner loop. Otherwise, EXIT inner loop.

    (c) Recompute $\tilde{g} \leftarrow \texttt{PrivateMed}(y_{i,1}, \ldots, y_{i,k})$, where `PrivateMed` is an $(\varepsilon_0, 0)$-differentially private algorithm for estimating the median of the data (see Theorem 2.6).

    (d) Output estimate $\tilde{g}$ and CONTINUE outer loop.

---

## 3 Differential Privacy as a Tool for Robust Streaming

In this section we present our main construction – algorithm `RobustSketch`. Recall that the main challenge when designing adversarially robust streaming algorithms is that the elements in the stream can depend on the internal state of the algorithm. To overcome this challenge, we protect the internal state of algorithm `RobustSketch` using differential privacy.

Suppose that we have an oblivious streaming algorithm $\mathcal{A}$ for a function $g$. In our construction we run $k$ independent copies of $\mathcal{A}$ with independent randomness, and feed the input stream to all of the copies. When a query comes, we aggregate the responses from the $k$ copies in a way that protects the internal randomness of each of the copies using differential privacy. In addition, assuming that the *flip number* [8] of the stream is small, we get that the number of times that we need to compute such an aggregated response is small. We use the sparse vector technique (algorithm `AboveThreshold`) [19] in order to identify the time steps in which we need to aggregate the responses of the $k$ copies of $\mathcal{A}$, and the aggregation itself is done using a differentially private algorithm for approximating the *median* of the responses.

In the next lemma we show that algorithm `RobustSketch` satisfies differential privacy w.r.t. the internal randomness of the different copies of $\mathcal{A}$. In our case, it suffices to guarantee differential privacy with a constant $\varepsilon$, and we fix $\varepsilon = \frac{1}{100}$.

**Lemma 3.1.** *Denote $\varepsilon = \frac{1}{100}$, and let $\delta \in (0,1)$ be a parameter. Algorithm `RobustSketch` satisfies $(\varepsilon, \delta)$-differential privacy (w.r.t. the collection of strings $R$).*

*Proof sketch.* Each execution of the outer loop consists of applying algorithm `AboveThreshold` and applying algorithm `PrivateMed`, each of which satisfies $(\varepsilon_0, 0)$-differential privacy. The lemma now follows from composition theorems for differential privacy (see Theorem 2.7). $\square$

Recall that algorithm `RobustSketch` might halt before the stream ends. In the following lemma we show that (w.h.p.) all the answers that `RobustSketch` returns before it halts are accurate. Afterwards, in Lemma 3.3, we show that (w.h.p.) the algorithm does not halt prematurely.

**Lemma 3.2.** *Let $\mathcal{A}$ be an oblivious streaming algorithm for a functionality $g$, that guarantees that with probability at least $9/10$, all its estimates are accurate to within multiplicative error of $(1 \pm \frac{\alpha}{10})$. Then, with probability at least $1 - \delta$ all the estimates returned by `RobustSketch` before it halts are accurate to within multiplicative error of $(1 \pm \alpha)$, even when the stream is chosen by an adaptive*

*adversary, provided that*

$$k = \Omega\left(\sqrt{\lambda \cdot \log\left(\frac{1}{\delta}\right)} \cdot \log\left(\frac{m}{\alpha\delta}\right)\right).$$

*Proof.* First observe that the algorithm samples at most $2m$ noises from the Laplace distribution with parameter $1/\varepsilon_0$ throughout the execution. By the properties of the Laplace distribution, with probability at least $1 - \delta$ it holds that *all* of these noises are at most $\frac{1}{\varepsilon_0}\log(\frac{2m}{\delta})$ in absolute value. We continue with the analysis assuming that this is the case.

For $i \in [m]$ let $\vec{a}_i = ((a_1, \Delta_1), \dots, (a_i, \Delta_i))$ denote the stream consisting of the first $i$ updates. Let $\mathcal{A}(r, \vec{a}_i)$ denote the estimate returned by the oblivious streaming algorithm $\mathcal{A}$ after the $i$th update, when it is executed with the random string $r$ and receives the stream $\vec{a}_i$. Note that $y_{i,j} = \mathcal{A}(r_j, \vec{a}_i)$. Consider the following function:

$$f_{\vec{a}_i}(r) = \mathbb{1}\left\{\mathcal{A}(r, \vec{a}_i) \in \left(1 \pm \frac{\alpha}{10}\right) \cdot g(\vec{a}_i)\right\}.$$

Observe that the function $f_{\vec{a}_i}(\cdot)$ is defined by $\vec{a}_i$. Recall that algorithm RobustSketch is $(\varepsilon = \frac{1}{100}, \delta)$-differentially private w.r.t. the collection of strings $R$ (see Lemma 3.1). Also recall that the updates in the stream $\vec{a}_i$ are chosen (by the adversary) by post-processing the estimates returned by RobustSketch. As differential privacy is closed under post-processing, we can view the *updates* in the stream $\vec{a}_i$, as well as the function $f_{\vec{a}_i}(\cdot)$, as the outcome of a differentially private computation on the collection of strings $R$. Therefore, by the generalization properties of differential privacy (see Theorem 2.8), assuming that $k \geq \frac{1}{\varepsilon^2}\log(\frac{2\varepsilon m}{\delta}) = \Theta\left(\log\frac{m}{\delta}\right)$, with probability at least $(1 - \frac{\delta}{\varepsilon}) = 1 - O(\delta)$, for every $i \in [m]$ it holds that

$$\left|\mathbb{E}_r[f_{\vec{a}_i}(r)] - \frac{1}{k}\sum_{j=1}^{k} f_{\vec{a}_i}(r_j)\right| \leq 10\varepsilon = \frac{1}{10}.$$

We continue with the analysis assuming that this is the case. Now observe that $\mathbb{E}_r[f_{\vec{a}_i}(r)] \geq 9/10$ by the utility guarantees of $\mathcal{A}$ (because when the stream is fixed its answers are accurate to within multiplicative error of $(1 \pm \frac{\alpha}{10})$ with probability at least $9/10$). Thus, for at least $(\frac{9}{10} - 10\varepsilon)k = 4k/5$ of the executions of $\mathcal{A}$ we have that $f_{\vec{a}_i}(r_j) = 1$, which means that $y_{i,j} \in (1 \pm \frac{\alpha}{10}) \cdot g(\vec{a}_i)$. That is, in every time step $i \in [m]$ we have that at least $4k/5$ of the $y_{i,j}$'s satisfy $y_{i,j} \in (1 \pm \frac{\alpha}{10}) \cdot g(\vec{a}_i)$.

**Case (a)** If the algorithm outputs an estimate on Step 3(b)iii, then, by our assumption on the noise magnitude we have that

$$\left|\left\{j : \tilde{g} \in \left(1 \pm \frac{\alpha}{2}\right) \cdot y_{i,j}\right\}\right| \geq \frac{k}{2} - \frac{2}{\varepsilon_0}\log\left(\frac{2m}{\delta}\right) \geq \frac{4k}{10},$$

where the last inequality follows by asserting that

$$k = \Omega\left(\frac{1}{\varepsilon_0}\log\frac{m}{\delta}\right) = \Omega\left(\sqrt{\lambda \cdot \log\left(\frac{1}{\delta}\right)}\log\left(\frac{m}{\delta}\right)\right).$$

So, for at least $4k/5$ of the $y_{i,j}$'s we have that $y_{i,j} \in (1 \pm \frac{\alpha}{10}) \cdot g(\vec{a}_i)$, and for at least $4k/10$ of them we have that $\tilde{g} \in (1 \pm \frac{\alpha}{2}) \cdot y_{i,j}$. Therefore, there must exist an index $j$ that satisfies both conditions, in which case $\tilde{g} \in (1 \pm \alpha) \cdot g(\vec{a}_i)$, and the estimate we output is accurate.

**Case (b)** If the algorithm outputs an estimate on Step 3d, then it is computed using algorithm PrivateMed, which is executed on the database $(y_{i,1}, \dots, y_{i,k})$. By theorem 2.6, assuming that[6]

$$k = \Omega\left(\frac{1}{\varepsilon_0}\log\left(\frac{\lambda}{\alpha\delta}\log n\right)\right) = \Omega\left(\sqrt{\lambda \cdot \log\left(\frac{1}{\delta}\right)} \cdot \log\left(\frac{\lambda}{\alpha\delta}\log n\right)\right),$$

then with probability at least $1 - \delta/\lambda$ algorithm `PrivateMed` returns an approximate median $\tilde{g}$ for the estimates $y_{i,1}, \ldots, y_{i,k}$, satisfying

$$|\{j : y_{i,j} \geq \tilde{g}\}| \geq \frac{4k}{10} \qquad \text{and} \qquad |\{j : y_{i,j} \leq \tilde{g}\}| \geq \frac{4k}{10}.$$

Since $4k/5$ of the $y_{i,j}$'s satisfy $y_{i,j} \in (1 \pm \frac{\alpha}{10}) \cdot g(\vec{a}_i)$, such an approximate median must also be in the range $(1 \pm \frac{\alpha}{10}) \cdot g(\vec{a}_i)$. This holds simultaneously for all the estimates computed in Step 3d with probability at least $1 - \delta$. Note that in Case (b) our estimate is actually accurate to within $(1 \pm \frac{\alpha}{10})$ rather than $(1 \pm \alpha)$.

Overall, with probability at least $1 - O(\delta)$, all the estimates returned by the algorithm are accurate to within a multiplicative error of $(1 \pm \alpha)$. □

We now show that, with high probability, the algorithm does not halt before the stream ends.

**Lemma 3.3.** *Let algorithm* `RobustSketch` *be executed with a parameter* $\lambda > \lambda_{\alpha/10,m}(g)$. *With probability at least* $1 - \delta$, *the algorithm does not halt before the stream ends.*

*Proof.* As in the proof of Lemma 3.2, with probability at least $1 - \delta$ it holds that

1. All of the Laplace noises sampled throughout the execution are at most $\frac{1}{\varepsilon_0} \log(\frac{2m}{\delta})$ in absolute value,

2. All of the estimates returned on Step 3d are accurate to within a multiplicative error of $(1 \pm \frac{\alpha}{10})$,

3. In every time step $i \in [m]$ we have that at least $4k/5$ of the $y_{i,j}$'s satisfy $y_{i,j} \in (1 \pm \frac{\alpha}{10}) \cdot g(\vec{a}_i)$.

We continue with the proof assuming that these statements hold. For $i \in [m]$ let $\tilde{g}_i$ denote the $i$th estimate that we output. Let $i_1 < i_2 \in [m]$ denote sequential time steps in which the algorithm outputs an estimate on Step 3d (and such that between $i_1$ and $i_2$ we compute the estimation using Step 3(b)iii). Since we do not change our estimate between time steps $i_1$ and $i_2$, we know that $\tilde{g}_{i_2-1} = \tilde{g}_{i_1}$.

Now, since in time step $i_2$ we exit the inner loop (in order to output the estimate using Step 3d), it holds that

$$\left| \left\{ j : \tilde{g}_{i_2-1} \notin \left(1 \pm \frac{\alpha}{2}\right) \cdot y_{i_2,j} \right\} \right| \geq \frac{4k}{10}.$$

Since at least $4k/5$ of the $y_{i_2,j}$'s satisfy $y_{i_2,j} \in (1 \pm \frac{\alpha}{10}) \cdot g(\vec{a}_{i_2})$, there must exist a $y_{i_2,j}$ such that $\tilde{g}_{i_2-1} \notin (1 \pm \frac{\alpha}{2}) \cdot y_{i_2,j}$ and $y_{i_2,j} \in (1 \pm \frac{\alpha}{10}) \cdot g(\vec{a}_{i_2})$. Hence, $\tilde{g}_{i_2-1} \notin (1 \pm \frac{\alpha}{4}) \cdot g(\vec{a}_{i_2})$.

Now recall that since in time step $i_1$ we return the estimate $\tilde{g}_{i_1} = \tilde{g}_{i_2-1}$ using Step 3d, it holds that $\tilde{g}_{i_1} = \tilde{g}_{i_2-1} \in (1 \pm \frac{\alpha}{10}) \cdot g(\vec{a}_{i_1})$. So, we have established that $\tilde{g}_{i_2-1} \notin (1 \pm \frac{\alpha}{4}) \cdot g(\vec{a}_{i_2})$ and that $\tilde{g}_{i_2-1} \in (1 \pm \frac{\alpha}{10}) \cdot g(\vec{a}_{i_1})$, which means that

$$g(\vec{a}_{i_2}) \notin \left(1 \pm \frac{\alpha}{10}\right) \cdot g(\vec{a}_{i_1}).$$

This means that every time we recompute $\tilde{g}$ on Step 3d, it holds that the true value of $g$ has changed by a multiplicative factor larger than $(1 + \frac{\alpha}{10})$ or smaller than $(1 - \frac{\alpha}{10})$. In that case, the number of times we recompute $\tilde{g}$ on Step 3d cannot be bigger than $\lambda_{\alpha/10,m}(g)$. Thus, if the algorithm is executed with a parameter $\lambda > \lambda_{\alpha/10,m}(g)$, then (w.h.p.) the algorithm does not halt before the stream ends. □

The next theorem is obtained by combining Lemma 3.2 and Lemma 3.3.

**Theorem 3.4.** *Let* $\mathcal{A}$ *be an oblivious streaming algorithm for a functionality* $g$, *that uses space* $L(\frac{\alpha}{10}, \frac{1}{10})$ *and guarantees accuracy* $\frac{\alpha}{10}$ *with success probability* $\frac{9}{10}$ *for streams of length* $m$. *Then there exists an adversarially robust streaming algorithm for* $g$ *that guarantees accuracy* $\alpha$ *with success probability* $1 - \delta$ *for streams of length* $m$ *using space*

$$O\left( L\left(\frac{\alpha}{10}, \frac{1}{10}\right) \cdot \sqrt{\lambda_{\frac{\alpha}{10},m}(g) \cdot \log\left(\frac{1}{\delta}\right)} \cdot \log\left(\frac{m}{\alpha\delta}\right) \right).$$

# 4 Applications

Our algorithm can be applied to a wide range of streaming problems, such as estimating frequency moments, counting the number of distinct elements in the stream, identifying heavy-hitters in the stream, estimating the median of the stream, entropy estimation, and more. As an example, we now state the resulting bounds for $F_2$ estimation.

**Definition 4.1.** *The* frequency vector *of a stream* $(a_1, \Delta_1), \ldots, (a_m, \Delta_m)$, *where* $(a_i, \Delta_i) \in ([n] \times \mathbb{Z})$, *is the vector* $f \in \mathbb{R}^n$ *whose $\ell$th coordinate is*

$$f_\ell = \sum_{i:a_i=\ell} \Delta_i.$$

*We write $f^{(i)}$ to denote the frequency vector restricted to the first $i$ updates.*

In this section we focus on estimating $F_2$, the second moment of the frequency vector. That is, after every time step $i$ we want to output an estimation for

$$\|f^{(i)}\|_2^2 = \sum_{\ell=1}^n \left| f_\ell^{(i)} \right|^2.$$

We will use the following definition.

**Definition 4.2** ([29]). *Fix any $\tau \geq 1$. A data stream* $(a_1, \Delta_1), \ldots, (a_m, \Delta_m)$, *where* $(a_i, \Delta_i) \in [n] \times \{1, -1\}$, *is said to be an $F_2$ $\tau$-bounded deletion stream if at every time step $i \in [m]$ we have*

$$\|f^{(i)}\|_2^2 \geq \frac{1}{\tau} \cdot \|h^{(i)}\|_2^2,$$

*where $h$ is the frequency vector of the stream with updates $(a_i, |\Delta_i|)$.*

The following lemma relates the bounded deletion parameter $\tau$ to the flip number of the stream.

**Lemma 4.3** ([8]). *The $\lambda_{\alpha,m}(\| \cdot \|_2^2)$ flip number of a $\tau$-bounded deletion stream is at most $O\left(\frac{\tau}{\alpha^2} \log m\right)$.*

The following theorem is now obtained by applying algorithm `RobustSketch` with the oblivious algorithm of [30] that uses space $O\left(\frac{1}{\alpha^2} \log^2(\frac{m}{\delta})\right)$.

**Theorem 4.4.** *There is an adversarially robust $F_2$ estimation algorithm for $\tau$-bounded deletion streams of length $m$ that guarantees $\alpha$ accuracy with probability at least $1 - \frac{1}{m}$. The space used by the algorithm is*

$$O\left( \frac{\sqrt{\tau}}{\alpha^3} \cdot \log^4(m) \right).$$

In contrast, the $F_2$ estimation algorithm of [8] for $\tau$-bounded deletion streams uses space $O\left(\frac{\tau}{\alpha^4} \cdot \log^3(n)\right)$. Specifically, the space bound of [8] grows as $\frac{\tau}{\alpha^4}$, whereas ours only grows as $\frac{\sqrt{\tau}}{\alpha^3}$ (at the cost of additional $\log(m)$ factors). As we mentioned, our results are also meaningful for the insertion-only model. Specifically,

**Lemma 4.5** ([8]). *The $\lambda_{\alpha,m}(\| \cdot \|_2^2)$ flip number of an insertion-only stream is at most $O\left(\frac{1}{\alpha} \log m\right)$.*

The following theorem is obtained by applying algorithm `RobustSketch` with the oblivious algorithm of [10] that uses space $\tilde{O}\left(\frac{1}{\alpha^2} \log(m) \log(\frac{1}{\delta})\right)$.

**Theorem 4.6.** *There is an adversarially robust $F_2$ estimation algorithm for insertion-only streams of length $m$ that guarantees $\alpha$ accuracy with probability at least $1 - \frac{1}{m}$. The space used by the algorithm is*

$$\tilde{O}\left( \frac{1}{\alpha^{2.5}} \cdot \log^4(m) \right).$$

In contrast, the $F_2$ estimation algorithm of [8] for insertion-only streams uses space $\tilde{O}\left(\frac{1}{\alpha^3} \cdot \log^2(m)\right)$. Our bound, therefore, improves the space dependency on $\alpha$ (at the cost of additional logarithmic factors).

## Broader Impact

Our work applies differential privacy and generalization bounds to make streaming algorithms robust to adversarial attacks and feedback loops (in which the value reported by the algorithm affects future updates). The idea of using differential privacy as a tool to protect against adversarial attacks on the randomness of the algorithm may be applicable more generally, when a randomized ML model that reports continuously is exposed to a dangerous feedback loop or malicious users. We believe that the connection we establish in this work is only the beginning, and that, following our work, ideas from the literature of differential privacy will continue to find new applications in the field of robust streaming and other related areas.

## Acknowledgments and Disclosure of Funding

The authors are grateful to Amos Beimel and Edith Cohen for many helpful discussions.

Haim Kaplan is partially supported by Israel Science Foundation (grant 1595/19), German-Israeli Foundation (grant 1367/2017), and the Blavatnik Family Foundation. Yishay Mansour has received funding from the European Research Council (ERC) under the European Union's Horizon 2020 research and innovation program (grant agreement No. 882396), and by the Israel Science Foundation (grant number 993/17). Uri Stemmer is partially supported by the Israel Science Foundation (grant 1871/19), and by the Cyber Security Research Center at Ben-Gurion University of the Negev.

## Footnotes

[6]We assume that the estimates that $\mathcal{A}$ returns are in the range $[-n^c, -1/n^c] \cup \{0\} \cup [1/n^c, n^c]$ for some constant $c > 0$. In addition, before running PrivateMed we may round each $y_{i,j}$ to its nearest power of $(1 + \frac{\alpha}{10})$, which has only a small effect on the error. There are at most $X = O(\frac{1}{\alpha}\log n)$ possible powers of $(1 + \frac{\alpha}{10})$ in that range, and hence, PrivateMed guarantees error at most $\Gamma = O(\frac{1}{\varepsilon_0}\log\left(\frac{\lambda}{\alpha\delta}\log n\right))$ with probability at least $1 - \delta/\lambda$. See Theorem 2.6.

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
