[Reviews · NeurIPS 2020]

Review 1

Summary and Contributions: The paper proposes a new framework for adversarially robust streaming algorithms. Classical streaming algorithms are designed to work against an oblivious adversary, i.e. one usually thinks of fixing the stream in advance, and then running the streaming algorithm on it. However, in many applications where one would like to have small summaries of the input sequence of data items the output of such summaries (i.e., streaming algorithms) may influence the execution path of the entire system, thereby creating a feedback look, and hence making the input stream adaptive to the actions of the streaming algorithm. Classical streaming algorithms are not designed to operate in such circumstances, and this paper aims at fixing this deficiency. More precisely, recent work of Ben-Elizier et al has introduced a framework for adversarially robust algorithms, and this paper proposes a different one, which results in nontrivial improvements in parameters for several fundamental applications. The proposed framework is based on differential privacy. The idea is to maintain several independent copies of a given streaming algorithm, and use differentially private median estimators to compute the answer whenever the answer has changed nontrivially. This last condition is also verified using a differentially private estimation method. The analysis is based on several powerful tools from the differential privacy literature, including generalization properties of differentially private algorithms.

Strengths: The idea to use differential privacy to make streaming algorithms robust to adversarial inputs is very natural, and it is great that the authors fleshed it out. The improvement in parameters is nonitrivial -- the new framework goes from linear to only square root dependence on the flip number of the stream. The analysis is interesting.

Weaknesses: One weakness is the lack of lower bounds on the dependence of the space on the flip number of the stream (or, the parameter \tau) -- are there any provably limitations here? This could also be a strength though, since it suggests interesting future directions.

Correctness: I did not verify all the details, but I do not have any reasons to doubt correctness (the writing quality is quite good, and what I did verify was correct).

Clarity: The paper is rather well-written.

Relation to Prior Work: The paper features a very good overview of prior work.

Reproducibility: Yes

Additional Feedback:


Review 2

Summary and Contributions: This paper uses differential privacy to hide internal randomness, thereby providing a framework for streaming algorithms when the input can be from an adversary that adapts to previous outputs. The framework improves upon recent results of O. Ben-Eliezer, et. al. [PODS 2020] by 1/epsilon factors for applications such as norm estimation.

Strengths: The strength of the work is in the novelty of applying differential privacy to hide the randomness internal to streaming algorithms. I have not previously seen tools from differential privacy being used in streaming algorithms. Seems like a good fit for a theoretical conference such as ITCS. General framework that improves on very recent work.

Weaknesses: The paper focuses on improving 1/epsilon factors and is somewhat loose in their bounds with logs. Quantitatively, the 1/epsilon improvements are incremental, such as 1/eps^3 to 1/eps^{2.5} for F_2 estimation. Not entirely convinced the presentation and results are catered toward the machine learning audience.

Correctness: The claims and methods seem correct to me. The minor questions listed below do not affect the overall claims. No experiments are provided.

Clarity: I think the main audience is the streaming algorithms community, who would benefit from more intuition on why the differential privacy theorems apply. For example Lemma 3.1 is not referenced, but I think it is actually implicitly applied in the proof of Lemma 3.2 to convert the adversarial stream to an oblivious stream.

Relation to Prior Work: Relevant prior work is discussed.

Reproducibility: Yes

Additional Feedback: In the generalization property Theorem 2.7, why is there no dependence on log|X|, where |X| is the size of the universe of possible samples, as in Bassily et. al.? Similarly, why isn't the dependence on epsilon in Theorem 2.7 be 1/epsilon^3? Why can the generalization bound be used on the indicator variables in the proof of Lemma 3.2? Is it because of post-processing? =======Post-rebuttal update======= I have increased my evaluation of the overall score and the confidence score by 1 for the following reasons: 1. The rebuttal emphasized that for turnstile streams, the reduction of the dependency of the flip number to sqrt(lambda) gives sublinear space algorithms for a larger range of turnstile streams, e.g., when the stream length is linear in the size of the universe for norm estimation. 2. The rebuttal addressed my concerns of correctness, which were minor to begin with and would not have affected asymptotics. 3. The rebuttal observed that the paper's applications to adversarial attacks and adaptive data analysis could indeed be of interest to the general ML community.


Review 3

Summary and Contributions: Summary and contributions: Briefly summarize the paper and its contributions. * (visible to authors during feedback, visible to authors after notification, visible to other reviewers, visible to meta-reviewers) This paper uses techniques from adaptive data analysis , in particular differential privacy, to give theoretical guarantees for streaming algorithms against adaptive adversaries in the turnstile model (with deletions allowed). In some parameter regimes they also improve over existing results in the oblivious insertion only setting. In particular their space requirements scale with sqrt(flip number) of the function being estimated, rather than linearly, at the cost of log factors. While the connection between adaptive adaptive data analysis and differential privacy is well understood, this appears to be the first use of differential privacy to protect the internal state of a randomized algorithm, which in addition to the practical importance of the results is an interesting theoretical contribution. The actual algorithm is conceptually simple, and uses a sub-sample and aggregate scheme that has been applied many times.

Strengths: - Claims are correct, paper is well-written - Contribution is novel due to the application of differential privacy in this context, and improvement over prior work - Very relevant to the community

Weaknesses: No major weaknesses. This is a strong submission.

Correctness: Yes

Clarity: Yes

Relation to Prior Work: Yes

Reproducibility: Yes

Additional Feedback: Typo in lemma 3.2: functionality -> function


Review 4

Summary and Contributions: The paper considers the adversarially robust streaming algorithms in the turnstile model. In this adversarial model, the adversary is adaptive, who feeds updates to the stream based on the previous inputs and the query answers outputted by the streaming algorithm. The most relevant work is due to Ben Eliezer et al., who considered the adversarially robust streaming algorithm in the insertion-only model, which can be generalized to limited deletions. Here, the goal is to output a (1+eps)-approximation to some desired statistics. The general framework is to turn oblivious streaming algorithms to adversarially robust algorithms. To do this, they defined a notion of the “flip number”, which is the number of times that the true answer changes by a factor of more than 1+eps during the stream, and then showed that by paying an extra factor of the flip number, one can construct an adversarially robust (1+eps)-approximation algorithm from the oblivious (1+eps/10)-approximation algorithm. This paper improves this extra factor to sqrt(flip number) with techniques from differential privacy (largely the sparse vector technique) but requires an additional log factor of the length of the stream. Therefore, it improves the dependence on eps for adversarially robust algorithms but may result in more log factors of the length of the stream.

Strengths: Improves the dependence of flip number in the framework and thus the eps (approximation parameter) dependence in the final algorithm. This is a good theoretical contribution, and is also useful for the streaming community or anyone who uses streaming algorithms in practice.

Weaknesses: For constant eps, it is not clear how good the improvement is as it also introduces additional log factor on the length of the stream. If eps = 0.1, a stream of length 2^10 (which is not long) would cancel out the improved dependence on 1/eps.

Correctness: The proofs look correct to me, though it is possible that I missed some gap.

Clarity: The paper is very well-written. A small typo: Line 164, unbounded -> is unbounded

Relation to Prior Work: The technique makes it look very different from the previous work.

Reproducibility: Yes

Additional Feedback: I think the paper has a solid contribution and should be published. [The author's response adequately addressed my earlier concern that NeurIPS might not be the right venue.]

[Author Response · NeurIPS 2020]

**Response to NeurIPS 2020 Reviews     #2822** *Adversarially Robust Streaming Algorithms via Differential Privacy*

We thank all four reviewers for their time and comments. Their suggestions will help us clarify the contributions of our work as we incorporate them in the next revision of our paper.

We view our work as providing both technical and conceptual contributions. The technical contribution is that we are the first to present adversarially robust streaming algorithms that work in the general turnstile model (where both positive and negative updates are allowed). Specifically, the previous work of Ben-Eliezer et al had space complexity that grows linearly with $\lambda$ (the flip number). In the turnstile model, $\lambda$ can be as big as the length of the stream, and hence the algorithms of Ben-Eliezer et al do not provide meaningful (worst-case) bounds. In contrast, our space complexity only grows as $\sqrt{\lambda}$, and hence, our algorithm has sublinear space also in the general turnstile model.

The conceptual contribution of our work is that it formally connects the fields of adaptive data analysis and differential privacy with the field of adversarially robust streaming. In particular, to the best of our knowledge, we are the first to use differential privacy in order to protect the internal randomness of the algorithm.

In the following, we respond to several specific points raised by the reviewers.

**Reviewer #1:** *"One weakness is the lack of lower bounds"*

We agree that our work does not close the door on this question. Our work is the first to show sub-linear space in the turnstile model for adversarially robust streaming, and suggests interesting future directions – in both upper and lower bounds.

**Reviewer #2:** *"benefit from more intuition on why the differential privacy theorems apply"*
*"In Theorem 2.7, why is there no dependence on $\log |X|$ and on $1/\varepsilon^3$ as in Bassily et al?"*

We will improve the presentation w.r.t. introducing differential privacy. Theorem 2.7 appears in Bassily et al as Theorem 7.2 (see their arXiv version).

**Reviewer #2:** *"Why can the generalization bound be used in the proof of Lemma 3.2? Is it because of post-processing?"*

Yes, it's because of post-processing. We will make it clear in the next revision of our paper.

**Reviewer #2 and #4:** *"The paper focuses on improving 1/epsilon factors"*
*"Quantitatively, the 1/epsilon improvements are incremental"*
*"For constant eps, it is not clear how good the improvement is"*

Our technical focus is on improving the dependency in $\lambda$ (the flip number) from linear to square root, which allows us to present the first adversarially robust algorithm for the general turnstile model (see paragraph at the beginning of this rebuttal). In the turnstile model we could have that $\lambda = \Theta(m)$, where m is the length of the stream, at which case previous works do not obtain meaningful bounds (even when the approximation parameter is constant). We use space at most roughly $\sqrt{\lambda}$, which is sublinear, and therefore obtain the first algorithm for the turnstile case. As a by-product, in some parameter regimes, we also improve over existing results in the insertion-only model (in terms of the approximation parameter), but this is not our focus.

**Reviewer #2 and #4:** *"Not entirely convinced the presentation and results are catered toward the ML audience"*
*"NeurIPS may not be the right venue"*

Our work applies differential privacy and generalization bounds to make streaming algorithms robust to adversarial attacks and feedback loops (in which the value reported by the algorithm affects future updates). Each of these topics, namely, differential privacy, generalization, adversarial attacks, and streaming and sketching, has been of interest to the ML community and addressed in previous NeurIPS conferences. In particular, our work has a large intersection with the field of adaptive data analysis, which is one of the areas that appeared in the call for papers (under Algorithms).

Our idea of using privacy as a tool to protect against adversarial attacks on the randomness of the algorithm may be applicable whenever a randomized ML model that reports continuously is exposed to a dangerous feedback loop or malicious users. We will make this connection more explicit in the next revision of our paper.

[Meta-Review · NeurIPS 2020]

All the reviewers really liked the connection between streaming algorithms and differential privacy, which was nice to see formally fleshed out and give a new square root dependence on the flip number, allowing for sublinear space for much longer streams. There are no complaints.